# Impact of COVID-19 on Dental Emergency Services in Cluj-Napoca Metropolitan Area: A Cross-Sectional Study

**DOI:** 10.3390/ijerph17217716

**Published:** 2020-10-22

**Authors:** Nausica Bianca Petrescu, Ovidiu Aghiorghiesei, Anca Stefania Mesaros, Ondine Patricia Lucaciu, Cristian Mihail Dinu, Radu Septimiu Campian, Marius Negucioiu

**Affiliations:** 1Department of Oral Health, University of Medicine and Pharmacy “Iuliu Hatieganu”, 400012 Cluj-Napoca, Romania; nausica_petrescu@yahoo.com (N.B.P.); rcampian@email.com (R.S.C.); 2Department of Dental Propaedeutic, University of Medicine and Pharmacy “Iuliu Hatieganu”, 400006 Cluj-Napoca, Romania; mesaros.anca@umfcluj.ro; 3Department of Maxillofacial Surgery and Implantology, University of Medicine and Pharmacy “Iuliu Hatieganu”, 400029 Cluj-Napoca, Romania; dinu_christian@yahoo.com; 4Department of Prosthetic Dentistry, University of Medicine and Pharmacy “Iuliu Hatieganu”, 400006 Cluj-Napoca, Romania; marius.negucioiu@umfcluj.ro

**Keywords:** COVID-19, emergency dental services, SARS-CoV-2, epidemiology, public health

## Abstract

This study aimed to assess dental care needs in Cluj region during the State of Emergency, declared due to the COVID-19 pandemic, compared to the same period of the year 2019. A descriptive retrospective analysis was conducted, which retrieved patients seeking emergency dental services at the Emergency Department of County General Hospital and “Iuliu Hațieganu” University of Medicine and Pharmacy, Cluj-Napoca, Romania, the only dental service available in April 2020. Recorded data cover the month of April 2020 and is compared with the same period of 2019. During April 2020, 724 patients received dental care, whereas only 160 patients were treated in April 2019 in the same facility. The number of patients with acute apical periodontitis, abscess, and acute pulpitis was significantly higher in 2020. The percentage of patients receiving sedative filling for the treatment of acute pulpitis in 2020 was significantly higher than in 2019, while the proportion of patients receiving pulpectomy was higher in 2019. The percentage of patients receiving endodontic drainage for the treatment of acute periapical periodontitis in 2020 was higher. This study demonstrates that the COVID-19 pandemic impacted the use of medical care services and could further impact the oral health and quality of life of patients.

## 1. Introduction

Since December 2019, various changes took place worldwide, due to COVID-19. It is known that COVID-19 is a respiratory disease caused by the virus: Severe acute respiratory syndrome coronavirus 2 (SARS-CoV-2) [1].

After SARS-CoV-2 occurrence in Wuhan, China [2], until present times, COVID-19 has spread globally, causing more than 868,422 of deaths. More than 26,238,508 people tested positive for SARS-CoV-2 and the number of people affected by the disease is ever increasing [3]. On 11 March 2020, the World Health Organization (WHO) declared that the COVID-19 situation can qualify as a global pandemic because of its fast spread worldwide [4].

Romania announced a State of Emergency on March 18th, through a Military Ordinance [5], which consisted in several restrictions: closing the schools, universities and research institutes, canceling public events, closing the churches and canceling religious events, isolation and quarantine of the infected, social isolation of all people [6]. In those circumstances, the dental offices were officially closed, only few selected dental care centers throughout each county remained open for emergencies. The emergency dental treatments could be given only with the appropriate protective equipment [7], because dentistry staff are the branch most exposed to COVID-19 from the entire medical field [8,9], given the fact that many dentistry procedures involve production of aerosols and COVID-19’s main path of transmission is through saliva droplets [10,11].

Cluj-Napoca metropolitan area consists of 1 major city (Cluj-Napoca) and the 19 surrounding communities totaling a population of 415,771 people and covering 1603 square kilometers. However, the census of the population is done only for people having their permanent residency, and in normal times, the population is increased by 13–15% with students, as Cluj-Napoca is one of Romania’s most important university centers.

The total number of dental offices in Cluj-Napoca, in December 2019, was of 1014. From those, only the Emergency Department of County General Hospital and “Iuliu Hațieganu” University of Medicine and Pharmacy dental setting could receive emergencies after the pandemic was declared for the period March 22 April 30. The County Clinical Hospital declared two locations open for dental emergencies: one for the general population, where non-COVID-19 patients were treated, and one for patients that were suspects of having been infected with COVID-19 or had positive results for COVID-19 infection testing. The Romanian College of Dentists adopted the plan of measures for performing emergency dentistry interventions (17/3/BExN/2020), in which only the following pathologies were considered emergencies:Post-extractional hemorrhage;Pain due to acute pulpitis;Pain due to acute apical periodontitis;Pericoronitis of the impacted teeth;Post-extractional alveolitis;Cellulitis/abscesses;Mandibular fractures;Temporomandibular joint dislocation;Dento-alveolar traumas (dislocations, avulsions, dental fractures with the impairment of the pulp chamber);Ulceronecrotic gingivostomatitis [12].

A strict protocol was applied in the dental care service: the first step was the phone triage, general health triage before entering the dental office and then dental triage. On the phone questionnaire, the patients were asked if they presented COVID-19 symptoms or had any contact with individuals who returned from other countries severely affected by COVID-19 and they were also asked about their dental symptoms, in order to determine if they were truly in need of an emergency treatment. If the dental problem could not be treated or postponed telephonically, through guidance or medication, the patient was invited to the emergency dental office. If the patient was considered to be a suspect of COVID-19 infection or had previously tested positive for COVID-19 infection they were redirected to a secondary location which was prepared to provide dental care for infected patients.

In light of the major changes, the present study aimed to assess the need of dental treatment in Cluj region during the State of Emergency declared due to the COVID-19 pandemic, compared to the same period of the year 2019.

## 2. Materials and Methods

A descriptive retrospective analysis was conducted in the present study which retrieved patients seeking emergency dental services in a public dental setting at the Emergency Department of County General Hospital and “Iuliu Hațieganu” University of Medicine and Pharmacy dental setting in April 2020 compared to April 2019. This dental setting was the main one from Cluj Metropolitan Area as it is placed in Cluj-Napoca, the major city of Cluj County. It was set to receive patients that were not considered to be a suspect of being infected with the SARS-CoV-2 virus, nor had previous positive COVID-19 tests.

The study received the approval of the Ethics Commission of “Iuliu Hațieganu” University of Medicine and Pharmacy, Cluj-Napoca (no. 132/24.03.2020). Recorded data cover the period from April 1st to May 1st, 2020 compared to April 1st to May 1st, 2019. April 2020 was the period of Romania’s State of Emergency when the private practices were not accredited to receive patients, only the Emergency Department of County General Hospital and “Iuliu Hațieganu” University of Medicine and Pharmacy dental setting was accredited to receive emergencies. However, in April 2019, the same dental setting was providing dental treatments, together with another 1014 dental offices.

All patients who received emergency dental services in the previously mentioned institution, between April 1st and May 1st, 2019 and 2020, were included in the study. Information about demographic characteristics (gender, age), dental pathology, and treatment was recorded and introduced into an Excel sheet. SPSS 20.0 was used for the statistical analysis: descriptive statistics and chi-square calculator.

## 3. Results

### 3.1. General Information

In April 2019, 160 patients visited the Emergency Department of County General Hospital and “Iuliu Hațieganu” University of Medicine and Pharmacy, whereas in April 2020, 2131 patients sought dental treatment in this department, as other dental emergency centers and private practices were closed by the State of Emergency dispositions. In April 2020, patients were triaged on the phone; this procedure involved 2131 patients. From those patients, 1400 were postponed or they had a dental problem that could be dealt with telephonically; 7 were redirected to the emergency service of oro-maxillo-facial surgery, because they were suspected to be SARS-CoV-2 positive; and 724 were treated in the secondary location Emergency Department of County General Hospital and “Iuliu Hațieganu” University of Medicine and Pharmacy. Gender distribution of the emergency cases from April 2019 and 2020 is represented in Figure 1.

Comparing the frequency of patients in the dental office, it was observed that during both periods of time, male patients needed dental treatment in a higher percentage than female patients. However, comparing the attendance of female patients to the dental office, it is to be noticed that a higher percentage (46.41%) sought dental treatments during the Emergency state, compared to 2019, whereas the percentage of male patients in need of dental treatment decreased from 2019 to 2020. During the State of Emergency, a higher percentage of male patients sought dental treatments (53.59%) than female patients. The only difference was recorded for the 20–29 years age group, where the percentage of female patients outnumbered the male patients.

The youngest patient seeking dental treatment in 2019/2020 was 2 years old, the median age of patients was 20 years in 2019 and 32 in 2020, and the eldest patients was 78 years old in 2019 and 96 years old in 2020. Statistically significant differences were recorded for the age groups 0–9 years (*X*^2^ = 25.9527, *p* = 0.001002) and 10–19 years (*X*^2^ = 9.6974, *p* = 0.001845), where the number of patients decreased during the Emergency state. No statistically significant differences were recorded between the other age groups.

The patients seeking emergency dental care during the Romanian National State of Emergency in April 2020 had a Gaussian distribution with regard to their age (Figure 2). Almost a quarter of the patients that were treated for dental emergencies, during the one-month period, were from the 20–29 age group (24.86%) and combined with the groups 30–39 (19.75%) and 40–49 (12.85%) comprise more than half of the patients. In most age groups, a relative equality between male and female patients was found, except in the 30–39 and 40–49 age groups, respectively, where male patients had more dental emergencies than women. During April 2019, the majority of the patients seeking dental treatments were in the 0–9 age group with 26.8%, 10–19 age group with 22.15%, and 20–29 age group with 38.48%.

Comparing the number of patients seeking dental treatments with the total number of COVID-19 confirmed cases and COVID-19-related declared deaths in Romania (Figure 3), data indicate that there is an indirect correlation between the number of patients attending dental emergency and the number of COVID-19-related declared deaths per age group.

While COVID-19 disease affected all ages, with a normal distribution, but mostly persons between 30 and 69 years of age, deaths due to COVID-19 were more frequent in the elder population of 50 years old and above [13,14,15,16,17]. In the same period of time, the age distribution of patients seeking dental treatments showed that more than 50% of them were between 20 and 49 years of age.

### 3.2. Disease Profile

Only nine diagnoses were considered dental emergencies by the Romanian College of Dentists (17/3/BExN/2020): post-extractional bleeding; pain due to acute pulpitis; pain due to acute apical periodontitis; pericoronitis of the impacted teeth; post-extractional alveolitis; cellulitis/abscesses; jaw/mandible fractures and dento-alveolar traumas (dislocations, avulsions, dental fractures with the impairment of the pulp chamber; dislocation of the temporo-mandibular joint;); ulceronecrotic gingivostomatitis. Table 1 indicates the distribution of disease type and diagnosis in the emergency department during April 2019 and April 2020. Besides these nine diagnoses, some other situations were also found among the reasons that determined the patients to seek dental treatments, such as: superficial and medium profound cavities, irritative lesions due to orthodontic appliance, tooth pathological mobility and prosthetic crown loosening.

In total, 9.49% of all patients from 2019, and 6.91% of the patients from 2020 registered at the Emergency Department of County General Hospital and “Iuliu Hațieganu” University of Medicine and Pharmacy, had two pathological conditions in need of treatment.

The most common emergency problems among patients during April 2019 and 2020 were acute pulpitis, acute apical periodontitis, abscess, and caries. The proportions of patients with acute apical periodontitis, abscess, and acute pulpitis were significantly higher in 2020 than in 2019 (Pearson chi-square test, X^2^ = 19.2139, *p* = 0.00012, X^2^ = 6.7688, *p* = 0.009277, X^2^ = 4.51, *p* = 0.0337). The proportion of caries was significantly higher in 2019 (X^2^ = 23.7975, *p* = 0.000168).

Dental pathology correlated with age group was also assessed for April 2020. Results are indicated in Table 2.

For patients seeking dental treatment during April 2020, it was observed that although acute pulpitis was present in all age groups, it mostly affected the age group between 20 and 39 years of age (51.64%). The same age population was also affected by acute apical periodontitis (45.74%). The age group 20–29 sought dental treatment for pericoronitis of impacted teeth: 81.82% of all pericoronitis diagnosis was found in this age group. They also presented the highest percentage of dento-alveolar traumas—36.36%. Forty percent of ulceronecrotic gingivitis was found in the 60–69-year-old population, but it was also present in 20% of the 2–9-year-old group, 10–19, and 20–29-year-old patients, respectively.

From the “Other” pathologies that led patients to seeking emergency treatments, 42.86% of injuries due to orthodontic appliances were found in the 30–39 age group with a higher prevalence in female patients, and 45.45% of pathological dental mobility was found in the 60–69 years of age group.

Comparing pathology distribution in gender groups for patients seeking dental treatment in April 2020, as indicated in Table 3, the higher incidence of acute pulpitis, acute apical periodontitis, and abscess was recorded in male patients. These pathologies were also the most frequent cause for female patients seeking treatment. Comparing female with male patients, pericoronitis, temporomandibular joint-dislocation, and caries were more frequent in female patients.

It was observed that in April 2019 there was no data entry for diagnosis such as pericoronitis of impacted teeth, dislocation of the temporo-mandibular joint, and irritation due to orthodontic appliances, and the most plausible cause for this is that such cases that had small incidence even in 2020 sought treatment in other dental offices and private practices.

### 3.3. Treatments for Common Emergency Conditions

The most frequent treatment performed for abscess was endodontic drainage both in 2020 as in 2019, with no statistically significant differences. The percentage of patients receiving sedative filling for the treatment of acute pulpitis in 2020 was significantly higher than that in 2019 (X^2^ = 23.421, *p* < 0.0001), while the proportion of patients receiving pulpectomy was higher in 2019 (X^2^ = 88.9291, *p* < 0.0001). The percentage of patients receiving endodontic drainage for the treatment of acute periapical periodontitis in 2020 was significantly higher than that in 2019 (X^2^ = 6.2504, *p* = 0.012417). The percentage of patients receiving temporary fillings for the treatment of caries in 2019 was significantly higher than that in 2020, but without any statistical significance, instead, the sedative filling was statistically significantly more often performed in the emergency status (X^2^ = 6.0076, *p* = 0.01422). Sedative filling was the most frequent treatment approach in April 2020 (Table 4).

## 4. Discussion

The present study represents a retrospective descriptive analysis of dental care services during one month of the state of emergency period due to the COVID-19 pandemic in Cluj-Napoca, Romania, in comparison with the same period of 2019. The presented data provide a very meaningful image about dental services during the time of the COVID-19 pandemic. Firstly, we observed a high increase in the attendance to the emergency dental services in the public setting, as it was the only one available. Previous to the pandemic outbreak, 1014 dental offices were providing dental care. At the beginning of the state of emergency, only two dental care provider sites were functioning, as they were the only ones accredited by the National Authorities. Therefore, during the reported period, all dental patients from the Cluj-Napoca Metropolitan area were treated in the public dental setting at the Emergency Department of County General Hospital and “Iuliu Hațieganu” University of Medicine and Pharmacy. As routine dental care was not available during the emergency state in Romania, the patients were only seeking emergency treatments, so an important shift in attendance to dental services was registered. COVID-19 significantly impacted people’s dental care seeking behavior in Cluj-Napoca, Romania, similar to other countries [18,19,20]. As the activity of the private dental offices in Romania was suspended, a significant increase in the number of patients seeking emergency treatment in the public setting was recorded. Although the availability of the emergency dental care services is convenient for the category of the population with difficulties in accessing dental care in private practices, no ”non-emergency” cases were registered. 

The limitation of this study lies in the fact that the increase in the number of patients is relative, as only the public dental setting at the County General Hospital and University of Medicine and Pharmacy ”Iuliu Hațieganu” was available. Although in this setting the number of patients increased, as in general there are more than 1014 facilities providing dental care in normal times, we presume that overall, there is a high decrease in the attendance of the patients to the dental settings. Additionally, in April 2020 only emergency treatments were available.

In the examined periods of time, the male patients sought emergency services in a higher percentage than women, although with slight differences between the two periods, which is in accordance with the data reported in other countries [18]. This might be correlated with a better compliance in women [21,22,23].

Age distribution for the population seeking treatment in 2020, as indicated in Figure 2, demonstrates that the 20–29 age group had the most dental emergencies followed by the 30–39-year group. Almost equal emergency incidence was registered in the 10–19 years group (13.93%) as for the 50–69-year groups (13.12%). This result indicates that people aged 20–50 years neglected their oral status, being in need of emergency care services in a higher percentage than the other age groups. The fact that elder people and children were not registered in a high percentage also shows that susceptible groups to various contagious respiratory diseases were probably fearing exposure to contamination [24]. This idea is also supported by the comparison of the number of deaths/infected persons and emergency room attendance per age group (Figure 3), which indicates that the higher the number of deaths per age group, the lower the dental treatment attendance.

The comparison with the situation from 2019 showed that in 2020 there was a more balanced distribution of patients with regard to age, as in 2019 there was an increased attendance of the younger population, probably because the population over 29 years old preferred private practices or other facilities that were open at the time. Furthermore, the population younger than 18 years old have dental treatments covered by the Romanian Health Insurance House, in a public dental setting.

Acute apical periodontitis (42.3%), acute pulpitis (33.3%), and cellulitis/abscess (9.3%) were the most frequent diagnosis during the State of Emergency in Cluj-Napoca, Romania, but were also the most common diagnostics in 2019, as indicated by other researchers [25,26,27]. Pulpitis and apical periodontitis were also the most frequent diagnosis of the patients from other two studies performed on UK patients presenting to an emergency center [28,29] and one study regarding solely endodontic emergencies of patients in Texas [30]. However, in Brazil, the most common reasons the patients made an appointment to the emergency service were dental pain, trauma, and broken restorations [31].

Our results correlate with those registered by Guo H. et al., 2020, and Bai J. et al., 2020, indicating similarities in dental problems worldwide. Some particularities for different age groups were registered. In the age group 10–19, the percentage of pulpitis exceeded that of apical acute periodontitis. This indicates that in this age group dental pathologies might be intercepted in early stages. In this group, irritations due to orthodontic appliances appear, indicating inclusion of this age group in orthodontic treatment. In the groups between 50 and 69 years, we found 72.72% of all pathological tooth mobility as the reason for emergency care attendance. As social activities were limited by authorities, the percentage of trauma was low, and 36.36% was encountered in the age group 20–29 years. Assessing dental pathology correlated to gender, the results of our study indicate higher percentages of acute pulpitis, acute apical periodontitis, and abscess in males, whereas female patients were diagnosed with pericoronitis, temporomandibular dislocation, and caries. Dental pathology registered in women could indicate a better oral status and more dental care [23]. As a recent publication suggested the correlation between oral vesiculobullous lesions and SARS-CoV-2 infection [32], it is interesting to observe that ulceronecrotic gingivitis was registered in high percentages in the 3–9-year-old group, 10–19-year-old group, 20–29-year-old group, and even more in the 60–69 years group. As COVID-19 patients can present intraoral manifestations, careful examination is of paramount importance.

The most frequent treatment performed for abscess was endodontic drainage, both in 2020 and in 2019, with no statistically significant differences. The percentage of patients receiving sedative filling for the treatment of acute pulpitis in 2020 was significantly higher than that in 2019 (X^2^ = 23.421, *p* < 0.0001), while the proportion of patients receiving pulpectomy was higher in 2019 (X^2^ = 88.9291, *p* < 0.0001). As pulpectomy is a dental procedure associated with high concentrations in aerosols, the limitation of this procedure can be associated with patients’ and dentists’ fear regarding the spread of COVID-19.

The percentage of patients receiving endodontic drainage for the treatment of acute periapical periodontitis in 2020 was significantly higher than that in 2019 (X^2^ = 6.2504, *p* = 0.012417). In 2019, the majority of patients diagnosed with acute periapical periodontitis received endodontic treatment. As the endodontic treatment is also associated with aerosols, the use of this therapeutic procedure was limited.

## 5. Conclusions

This study demonstrates that the COVID-19 pandemic impacted, besides the economic and social life of population, the use of medical care services. The main cause of attendance to the emergency dental office was acute apical periodontitis and acute pulpitis, whereas dental trauma incidence was low.

The results of this study are helpful for predicting future dental needs. In the post-COVID period, people’s requirements for dental care could grow. In light of this, implementing prevention and control measures could be of paramount importance for the oral health status of the population.

In the near future, supplementary measures for infection prevention have to be taken in the dental office, which can lead to increased costs for treatment, leading to a change of attendance to dental care services, thus impacting the oral health and the quality of life of the patients. Careful planning and management of dental healthcare are mandatory, in order to adapt to the new conditions.

## Figures and Tables

**Figure 1 ijerph-17-07716-f001:**
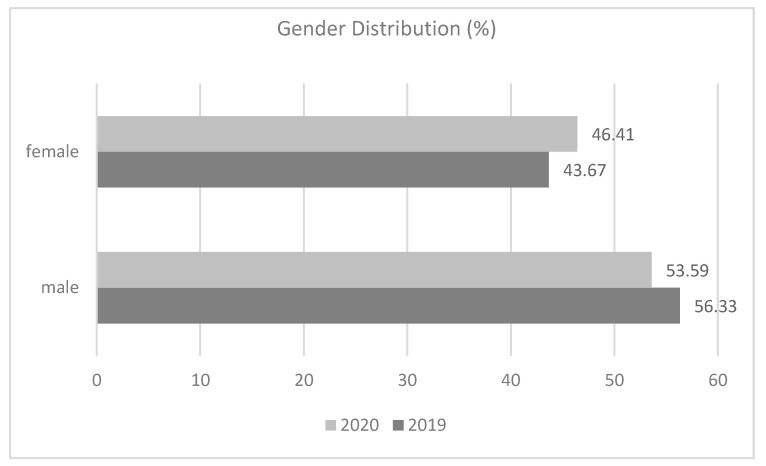
Differences in gender distribution in April 2019 compared to 2020.

**Figure 2 ijerph-17-07716-f002:**
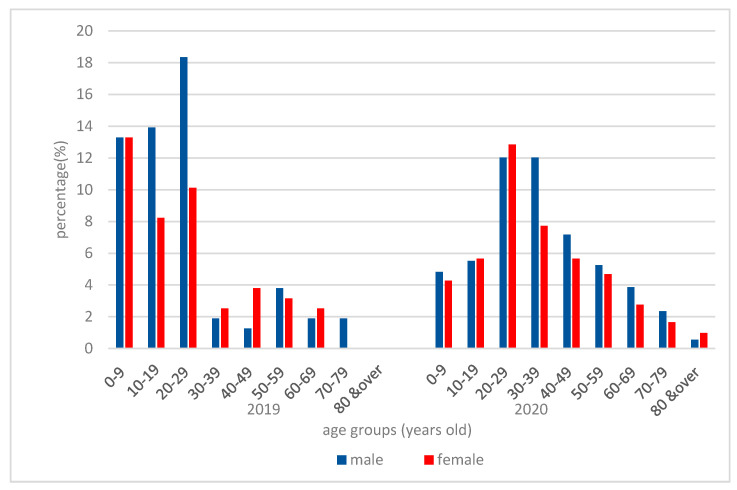
Comparison of patients’ age and gender distribution at the emergency room at the Emergency Department of County General Hospital and “Iuliu Hațieganu” University of Medicine and Pharmacy, in April 2019 and April 2020.

**Figure 3 ijerph-17-07716-f003:**
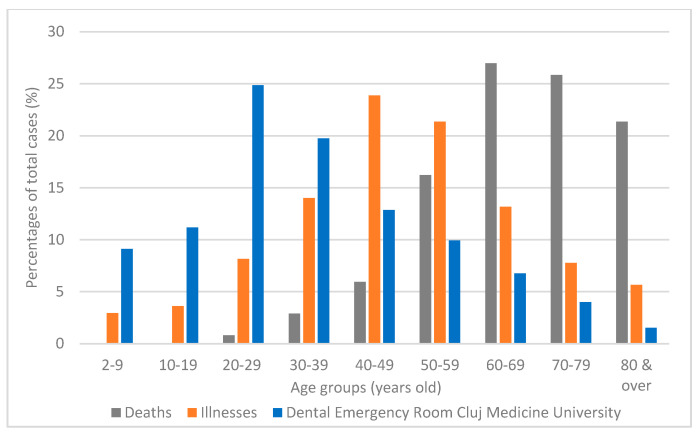
Comparison of the number of COVID-19 deaths/infected persons nationwide with the attendance in the Emergency Department of County General Hospital and “Iuliu Hațieganu” University of Medicine and Pharmacy Cluj Napoca per age group in the emergency period 1–30 April 2020.

**Table 1 ijerph-17-07716-t001:** Distribution of disease type and diagnosis in the emergency department during April 2019 and April 2020.

Pathology	No/2020	No/2019	Pathology	2020 (%)	2019 (%)
Acute pulpitis	244	35	Acute pulpitis	33.47	20.35
Acute apical periodontitis	324	29	Acute apical periodontitis	41.17	16.86
Pericoronitis of the impacted teeth	11		Pericoronitis of impacted teeth	1.4	0
Post-extractional alveolitis	1		Post-extractional alveolitis	0.13	0
Abscess	73	5	Abscess	9.28	2.91
Dislocation temporo-mandibular joint	1		Dislocation temporo-mandibular joint	0.13	0
Dento-alveolar trauma	40	2	Dento-alveolar trauma	5.08	1.16
Ulcer-necrotic gingivitis	6	7	Ulcer-necrotic gingivitis	0.76	4.07
Orthodontic appliance irritation injury	7		Orthodontic appliance irritation injury	0.89	0
Prosthetic crown loosening	12		Prosthetic crown loosening	1.52	0
Pathological dental mobility	12	2	Pathological dental mobility	1.52	1.16
Other injury (irritative, ulcerative, decubitus)	43	5	Other injury (irritative, ulcerative, decubitus)	5.46	2.91
Caries	13	87	Caries	1.65	50.58

**Table 2 ijerph-17-07716-t002:** Dental pathology distribution according to age groups during April 2020.

Percentage (%) Calculated from the Total Number of Each Diagnosis	2–9 Years	10–19 Years	20–29 Years	30–39 Years	40–49 Years	50–59 Years	60–69 Years	70–79 Years	>80 Years
Acute pulpitis	19 7.79%	34 13.93%	69 28.28%	57 23.36%	27 11.07%	22 9.02%	10 4.1%	5 2.05%	1 0.41%
Acute apical periodontitis	30 9.46%	28 8.83%	82 25.87%	63 19.87%	49 15.46%	27 8.52%	22 6.94%	12 3.79%	4 1.26%
Pericoronitis of the impacted teeth	0	1 9.09%	9 81.82%	1 9.09%					
Post-extractional alveolitis					1 (100%)				
Abscess	13 18.57%	7 10%	9 12.86%	13 18.57%	5 7.14%	13 18.57%	4 5.71%	5 7.14%	1 1.43%
Jaw/mandible fracture			1 (100%)						
Dento-alveolar traumas (fractures)	3 13.64%	1 4.55%	8 36.36%	3 13.64%	4 18.18%	2 9.09%		1 4.55%	
Ulcer-necrotic gingivitis	1 20%	1 20%	1 20%				2 40%		
**Other**									
Caries	3 23.08%	1 7.69%		4 30.77%	2 15.38%	1 7.69%	2 15.38%		
Orthodontic appliance irritation injury		1 14.29%	2 28.57%	3 42.86%			1 14.29%		
Other injury (irritative, ulcerative, decubitus)	3 17.65%	1 5.88%	2 11.76%	4 23.53%	1 5.88%	2 11.76%	1 5.88%	2 11.76%	1 5.88%
Pathological dental mobility		1 9.09%			1 9.09%	3 27.27%	5 45.45%		1 9.09%
Prosthetic crown loosening			3 27.27%		3 27.27%		3 27.27%	1 9.09%	1 9.09%

**Table 3 ijerph-17-07716-t003:** Dental pathology distribution according to gender for April 2020/2019.

Male/Female Comparison	2020 Male	2020 Female	2019 Male	2019 Female
Acute pulpitis	130 (32.67%)	111 (33.13%)	20 (20%)	15 (20.54%)
Acute apical periodontitis	174 (43.72%)	133 (39.71%)	16 (16%)	13 (17.81%)
Pericoronitis of the impacted teeth	3 (0.75%)	9 (2.68%)	0	0
Abscess	38 (9.55%)	29 (8.65%)	2 (2%)	3 (4.11%)
Dislocation temporo-mandibular joint	0	1(0.3%)	0	0
Dento-alveolar trauma	12 (3.02%)	13 (3.87%)	2 (2%)	1 (1.37%)
Ulcer-necrotic gingivostomatitis	3 (0.75%)	2 (0.6%)	6 (2%)	1 (1.37%)
**Other diagnosis of which**				
Caries	3 (0.76%)	9 (2.68%)	49 (49%)	38 (52.05%)
Prosthetic crown loosening	6 (1.51%)	5 (1.49%)	1 (1%)	1 (1.37%)
Orthodontic appliance irritation injury	4 (1.01%)	6 (1.79%)		
Other injury (irritative, ulcerative, decubitus)	6 (1.51%)	8 (2.4%)	4 (4%)	1 (1.37%)

**Table 4 ijerph-17-07716-t004:** Comparison of the treatment of common oral emergencies during April 2019 and April 2020.

Percentage of Total Therapeutic Procedures.	April 2020 (%)	April 2019 (%)
sedative filling	191 (29.28%)	1 (0.63%)
drainage(endodontic)	92 (14.09%)	33 (21.38%)
drainage/antiseptic lavage	75 (11.46%)	1 (0.63%)
drainage/sedative filling	6 (0.97%)	0
extraction	14 (2.07%)	6 (3.77%)
incision/drainage	8 (1.24%)	5 (3.14%)
incision/drainage/antiseptic lavage	3 (0.55%)	0
antiseptic lavage	12 (1.8%)	3 (1.89%)
pulpectomy	11 (1.66%)	52 (33.33%)
suppression of irritant factor	12 (1.8%)	0
filling		55 (35.64%)
other	31 (4.7%)	0
examination/consultation only	198 (30.36%)	0

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
