# Peer review of "Impact of COVID-19 on Dental Emergency Services in Cluj-Napoca Metropolitan Area: A Cross-Sectional Study"

_ijerph, 2020, doi:10.3390/ijerph17217716_

Round 1
Reviewer 1 Report
Thank you for the opportunity to review this study again following resubmission. The manuscript is vastly improved however there are still revisions required prior to acceptance.
Significance testing is reported for age, which statistical test did you use? There is no mention of any inferential statistics in your methodology.
Figure 2 needs tidying up, it's very difficult to look at with all the small numbers on the labels and the x axis is very messy. I'm assuming "yo" means "years old" but you need to state this abbreviation somewhere. If you state in your axis label that the unit is years you can delete the "yo" from each group to make it neater. You probably don't need the labels on the bars either as you can estimate them from the y axis.
Figure 3 there is a typing error in the title of the graph, again needs tidying up as is very messy. For all figures the title of the graph could go in the figure legend then you don't need a title and this gives you more room in the graph to neaten it up.
Line 157-160 where you talk about age groups of people dying, you need a reference to show where you got this information from.
Line 185, please move the chi square test values to brackets immediately following the diagnosis so it's clear which test relates to what diagnosis. Chi square should also be reported in the following format X2 (degress of freedom, N = sample size) = chi-square statistic value, p = p value.
Please note the correct spelling of pericoronitis.
Line 302 - this paragraph is just a repeat of the results and doesn't add anything to the discussion.
There are a wide range of studies on emergency dental care during the pandemic now published, I would expect to see more of these referenced in the discussion for comparison to your results.
Author Response
Dear Reviewer,
On behalf of all coauthors I would like to express our consideration for your involvement and for the suggestions you have made to the article we have submitted. Following these recommendations, we have made the modifications accordingly, hoping to improve our paper to your expectations. We have uploaded the revised manuscript with all the changes indicated with green.
Please find below the answer to your review.
With my best regards,
Corresponding author
Reviewer 1
Thank you for the opportunity to review this study again following resubmission. The manuscript is vastly improved however there are still revisions required prior to acceptance.
- Significance testing is reported for age, which statistical test did you use? There is no mention of any inferential statistics in your methodology.
Thank you for noticing that the name of the test was missing. We have mentioned the test that has been used in the materials and methods section.
- Figure 2 needs tidying up, it's very difficult to look at with all the small numbers on the labels and the x axis is very messy. I'm assuming "yo" means "years old" but you need to state this abbreviation somewhere. If you state in your axis label that the unit is years you can delete the "yo" from each group to make it neater. You probably don't need the labels on the bars either as you can estimate them from the y axis.
Thank you for your suggestion, we have improved Figure 2.
- Figure 3 there is a typing error in the title of the graph, again needs tidying up as is very messy. For all figures the title of the graph could go in the figure legend then you don't need a title and this gives you more room in the graph to neaten it up.
- and 3. We have done as suggested. The figures are more clear now.
- Line 157-160 where you talk about age groups of people dying, you need a reference to show where you got this information from.
We have added the reference for the the information source.
- Line 185, please move the chi square test values to brackets immediately following the diagnosis so it's clear which test relates to what diagnosis. Chi square should also be reported in the following format X2 (degress of freedom, N = sample size) = chi-square statistic value, p = p value.
We have done as suggested.
- Please note the correct spelling of pericoronitis.
We have corrected the spelling for pericoronitis.
- Line 302 - this paragraph is just a repeat of the results and doesn't add anything to the discussion.
We have deleted the mentioned phrase from the discussion.
- There are a wide range of studies on emergency dental care during the pandemic now published, I would expect to see more of these referenced in the discussion for comparison to your results.
In the discussion section we have added more references and compared our results with those of other recently published studies.

Reviewer 2 Report
Dear Authors
The manuscript is well modified and strengthened by the latest information. Read carefully the final manuscript after acceptance.
Author Response
Dear Reviewer,
On behalf of all coauthors I would like to express our consideration for your involvement and for the suggestions you have made to the article we have submitted. Following these recommendations, we have made the modifications accordingly, hoping to improve our paper to your expectations. We have uploaded the revised manuscript with all the changes indicated with green.
Please find below the answer to your review.
With my best regards,
Corresponding author
Reviewer 2
The manuscript is well modified and strengthened by the latest information. Read carefully the final manuscript after acceptance.
Thank you for your positive feedback.
